# Oral Health–Related Quality of Life and Missing Teeth in an Adult Population: A Cross-Sectional Study from Poland

**DOI:** 10.3390/ijerph19031626

**Published:** 2022-01-31

**Authors:** Ewa Rodakowska, Jacek Jamiolkowski, Joanna Baginska, Inga Kaminska, Katarzyna Gabiec, Zofia Stachurska, Marcin Kondraciuk, Marlena Dubatowka, Karol Adam Kaminski

**Affiliations:** 1Department of Clinical Dentistry-Cariology, University of Bergen, 5009 Bergen, Norway; 2Department of Population Medicine and Lifestyle Diseases Prevention, Medical University of Bialystok, 15-269 Bialystok, Poland; jacek.jamiolkowski@umb.edu.pl (J.J.); zofia.stachurska@umb.edu.pl (Z.S.); marcin.kondraciuk@umb.edu.pl (M.K.); marlena.dubatowka@umb.edu.pl (M.D.); karol.kaminski@umb.edu.pl (K.A.K.); 3Department of Dentistry Propaedeutics, Medical University of Bialystok, 15-295 Bialystok, Poland; joanna.baginska@umb.edu.pl; 4Department of Integrated Dentistry, Medical University of Bialystok,15-276 Bialystok, Poland; inga.kaminska@umb.edu.pl; 5Private Dental Clinic ’Lux-Dent’ Stomatologia, 15-668 Bialystok, Poland; gabieck@o2.pl

**Keywords:** missing teeth, tooth loss, populational study, oral health, oral health related quality of life

## Abstract

The aim of the study was to determine oral health–related quality of life (OHRQoL) using the measures Geriatric/General Oral Health Assessment (GOHAI) and Oral Health Impact Profile (OHIP-14) in relation to missing teeth in the Polish population aged 20–79. This was a cross-sectional study carried out among 1112 randomly selected participants. The mean age was 48.72 and mean number of teeth was 20.12. Altogether, in the GOHAI, the percentage that gave a positive response to each question ranged from 3.3% to 48.0%; in the OHIP-14, these answers ranged from 2.4% to 25.1%. The GOHAI measure was statistically significant, with more grouping variables than the OHIP-14 measure. Both measures showed significant associations with gender, age, dry mouth, education, professional status, number of teeth, and upper and lower total dentures. We detected a significant relationship between oral health–related quality of life and the factors influencing the presence or absence of dentition. Missing teeth were statistically associated with GOHAI, OHIP-14, advanced age, self-reported dry mouth, lower education, higher Body Mass Index (BMI), lower professional status, diabetes, myocardial infraction, and total dentures in upper or/and lower jaws. However, edentulous individuals had two times higher risk of having an OHIP-14 score above the median. This suggests that oral health practitioners should work to prevent oral diseases that lead to tooth loss in their patients, starting from an early age.

## 1. Introduction

Oral diseases can affect people around the world during their lifetime. Up to 3.5 billion people can be affected by oral diseases [1]. The World Health Organization (WHO) recognizes dental caries and periodontal disease, which are the main causes of tooth loss, as a significant public health burden [2,3]. According to the Global Burden of Disease 2017, untreated dental caries in permanent teeth is the most common health condition [4]. Oral health is an integrated part of general health, and the link between them seems to be unprecedented. A connection has been drawn between tooth loss and cardiovascular disease, diabetes, stroke, high blood pressure, respiratory diseases, dementia, and mortality [5,6,7,8,9,10]. On the other hand, though tooth loss and edentulism is decreasing in most European countries, it can still lead to deterioration of masticatory function and poor nutritional status [11,12]. When exposed to pain, problems with chewing, eating, and smiling as well as communication due to missing, fractured, or discolored teeth can occur, thus impacting social life, leading to lower self-esteem, and ultimately lowering quality of life [3]. In addition, dental treatment makes up 20% of out-of-pocket health expenditures in most high-income countries [13].

The situation in Poland regarding dental caries and the prevalence of missing teeth in the adult population has improved, but in comparison with other European countries, the above indicators remain high [14,15]. In Poland, the percentage of 35- to 44-year-old adults with full dentition between 2010 and 2017 increased from 24% to 43%. The frequency of dental caries decreased to 98.7%, but it remains at high and unsatisfactory levels. The percentage of edentulous senior citizens aged 65–74 also decreased from 40.8% to 18.8%, but only 27% had at least 20 teeth. The mean number of teeth increased from 6.1 in 2010 to 12.5 in 2019, but on average 3.5 teeth were diagnosed with dental caries. Epidemiological data on the prevalence and incidence of dental caries and tooth loss vary from country to country [11,12]. In Europe, the highest prevalence of edentulous senior citizens is in Bulgaria and Turkey (55% and 48%, respectively), and the lowest is in Sweden (2.8%). In the same age group, the DMFT (decay, missing, filled teeth) varied between 14.6 (Spain) and 25.8 (in Turkish citizens) [15].

Oral health–related quality of life (OHRQoL) is a multidimensional concept describing quality of life in relation to oral health and oral health diseases [16,17]. There is strong evidence that tooth loss is associated with impairment of OHRQoL. Moreover, this association seems to be independent from the instrument used for assessing OHRQoL [18]. The WHO identifies OHRQoL as an important element of the Global Oral Health Program [19]. Among many questionnaires describing OHRQoL, the two that are the most used are the Geriatric/General Oral Health Assessment Index (GOHAI) and the Oral Health Impact Profile (OHIP) [16,20]. They vary in the content of the questionnaires and present different sides of OHRQoL. The OHIP-14 places a greater focus on behavioral and psychological outcomes. On the other hand, the GOHAI focuses on oral pain/discomfort and functional limitations [21]. Initially, the OHIP consisted of 49 questions, which were then shortened by Slade and Spencer to 14 [22,23]. This version is currently used. It aims to assess seven dimensions of the impact of oral conditions on people’s OHRQoL. It includes functional limitation, physical pain, psychological discomfort, physical disability, psychological disability, social disability, and handicap [17,22]. This measure is centered on the patient, placing greater emphasis on more severe as well as less common issues that are psychological and behavioral [24]. Moreover, it better detects psychosocial effects within individuals and groups and better conforms to the main standards of OHRQoL according to Locker [24]. Another measure, the GOHAI, is a 12-item questionnaire, originally developed for use with the senior population. It evaluates the following dimensions of OHRQoL: physical and psychosocial functions and pain or discomfort. Physical elements include aspects as eating, speech, and swallowing. Psycho-social functioning is assessed through questions of worry, concerns about oral health, dissatisfaction with appearance, self-consciousness related to oral health, and avoidance of social contacts because of oral problems. The last dimension concerns medication use and oral discomfort [25]. The GOHAI refers to slight clinical changes and immediate clinical aspects reflecting subjective oral health status [24,26]. Both scales consider different time periods: three months for GOHAI and one year for OHIP-14.

This is the first study carried out on a such large scale in Poland with randomly selected participants. Owing to this study, we can determine the status of the general and oral health of the population of Bialystok city. These data can be used in the future, for example, to create more precise preventive programs. The aim of the study was to determine oral health–related quality of life (OHRQoL) using both the GOHAI and OHIP-14 in relation to missing teeth in the Polish population aged 20–79. Our hypothesis assumed that, based on the objective data of the age of the participants, oral health, and associate general diseases, we could expect poorer oral health–related quality of life.

## 2. Materials and Methods

### 2.1. Study Population

This was a cross-sectional study conducted between July 2017 and May 2021. It was approved by the Ethics Committee of the Medical University of Bialystok, Poland (R-I-002/108/2016) in conformity with Declaration of Helsinki. Participants were randomly selected based on the Bialystok mayor’s office database. As many as 3246 residents were invited to participate. A total of 1196 individuals responded to the invitation. Written consent was obtained before administration of questionnaire and dental examination. The final sample consisted of 1112 participants, as 84 people either refused to participate in the dental examination or did not fill in the whole questionnaire. The final response rate was 34.26%. Moreover, the minimal sample size calculation was determined based on the average number of adult residents of Bialystok city between 20 and 79 years old (207,676 residents for the year 2017), the percentage of people with at least one missing tooth (fraction size 50%), a maximum error of 5%, and a confidence level of 95%. The minimum sample size was set at 383 people. 

### 2.2. Data Acquisition

Participants completed a detailed questionnaire survey covering a range of questions about their socioeconomic status, current and past illnesses, and health habits. For the purpose of our research, variables studied included gender, age, education, place of residence, professional status, and medical history (diabetes mellitus, myocardial infraction, BMI—Body Mass Index, hypertension). Information on diseases was based on the interview questionnaire. BMI was assessed based on participants’ body weight and height measurements according to the formula BMI = body weight in kg/height in m^2^. To assess OHRQoL, the Polish versions of both GOHAI and OHIP-14 scales were used, which were both validated in a previous study by Rodakowska et al. [21]. The participants could choose one out of five answers on a 5-point Likert scale, from 0 for ‘never’ up to 4 for “always”. As the GOHAI consists of 12 questions, the maximum score is 48. The maximum score of the OHIP-14 is 56. Lower scores indicate better OHRQoL. In addition to the questionnaires, participants declared the occurrence of dry mouth (yes/no) and the presence or absence of complete dentures. 

Dental examination was conducted by four calibrated dentists in the dental examination room with professional units, illumination, and without use of a saliva ejector, air jet, or magnification glasses, according to WHO criteria [27]. Dental examination was carried in a half-mouth design on the right or left side, randomly chosen in alternate patients [28]. Wisdom teeth were excluded from the examination. In the survey, we used the CAST (Caries assessment Spectrum and Treatment) index to record dental caries, missing teeth, and filled teeth [29]. This index covers all aspects of caries, from nonactivated carious lesions to missing teeth, and allows for exclusion of DMF components. The D component reflected the codes 4,5,6, and 7, the M component reflected code 8, and the F component reflected code 2 in the CAST index. Based on these data, the number of remaining teeth, teeth with dental caries (D), missing teeth (M), and fillings (F) were calculated, and the decayed, missing, and filled teeth (DMFT) index was determined. 

### 2.3. Statistical Analysis 

The Kruskal–Wallis test and the Mann–Whitney U test were used to compare the GOHAI and the OHIP-14 scores in relation to age, gender, education, place of residence, professional status, diabetes mellites, myocardial infraction, BMI, and hypertension. Logistic regression models were used to assess strengths of association between oral health status, selected diseases, and OHIP-14 and GOHAI scores, dichotomized by median. Results were presented as odds ratio: 95% confidence intervals for odds ratios and p-values of statistical tests to verify the statistical hypothesis that OR ≠ 1. Spearman’s rank correlation coefficients were used to measure correlations between the OHIP-14 or GOHAI scores and age, DT, MT, and FT (decay teeth, missing teeth, filled teeth) as well as DMFT. The statistical analysis was performed using the IBM SPSS Statistics 27.0 software (IBM Corp. Released 2020. IBM SPSS Statistics for Windows, Version 27.0. IBM Corp, Armonk, NY, USA). Statistical hypotheses were verified with a significance level of 0.05.

## 3. Results

The final sample consisted of 1112 participants between 20 and 79 years old. Among them, 54.85% were women. Only 19.87% participants were over 65 years old. Mean age was 48.72. Substantial majorities were working (65.20%),, and 50.80% declared higher education. Furthermore, 65.91% of participants had over 20 teeth. On average, the number of teeth was 20.12.

Table 1 shows the percentage of participants who responded very often, fairly often, and occasionally to the GOHAI and OHIP-14 questions. Altogether, for the GOHAI, the percentage that gave a positive response to each question ranged from 3.3% to 48.0%. Participants had the fewest problems with uncomfortable swallowing and were most likely to be worried and concerned with problems with teeth/gums/dentures (42.0%) and sensitive teeth/gums to hot/cold (48.0%). For the OHIP-14, these answers ranged from 2.4% to 25.1%. This means that participants had the fewest problems with being unable to function and the most problems with eating foods because of their teeth, mouth, and dentures.

Table 2 presents mean values of the OHIP-14, GOHAI, DT, MT, FT, and DMFT as well as other variables. The GOHAI measure was statistically significant, with more grouping variables than the OHIP-14 measure. Both the GOHAI and OHIP-14 measures showed significant associations with gender, age, dry mouth, education, professional status, number of teeth, and upper and lower total dentures. Moreover, the GOHAI showed significant association with BMI, diabetes mellitus, and hypertension. MT, FT, and DMFT showed significant associations with all grouping variables, contrary to DT, which was significantly associated with gender, age, education, professional status, and having upper and lower total dentures.

Age, decayed, missing and filled teeth, and the DMFT (decayed, missing and filled teeth) index were significantly related to both the GOHAI and the OHIP14 (Table 3). 

The likelihood of having OHIP-14 and GOHAI scores above the median value in relation to oral health status and selected diseases is shown in Table 4. Number of teeth, total dentures in upper and/or lower jaw, and dry mouth were significantly associated with both scales. The higher number of preserved teeth, the lower the risk of obtaining an OHIP-14 and GOHAI score above the median. Individuals with total upper and lower dentures and dry mouth had between 1.5–2.5 times higher risk of having the OHIP-14 score above the median. Individuals with total upper and lower dentures and dry mouth had between 2.0–2.5 times higher risk of having the GOHAI score above the median. Additionally, edentulousness and myocardial infraction ware statistically significantly associated with OHIP-14. Diabetes and hypertension were not statistically associated with either scale describing OHRQoL. 

## 4. Discussion

The oral health–related quality of life in our study was not dissatisfying. Our study demonstrates that the number of teeth is a notable issue, not only in the elderly population with a small number of teeth, but also among adults aged around 50 who mostly have 20 teeth [21].

From what is known, this study is the first study that evaluates OHRQoL in the Polish population on such a large scale. An undeniable strength of the present study is the randomization of participating respondents, proper sample size calculation, wide age range, and unified dental examination conditions (dental unit, light). That is why our results can be generalized and compared with other results from different countries. The weakness of our study is the low response rate. However, the number of participants was three times the minimum sample size calculation. Another plausible limitation can be the long questionnaire, which could create a “fatigue effect” among participants, a phenomenon described in previous study [21].

We used two different scales for measuring OHRQoL in the present study. For objectively examining oral diseases, both instruments are well recognized measures that assess the oral health–related quality of life in adults and senior population. However, they differ in the item content and time reference. In our study, OHRQoL was high; the mean OHIP-14 was 2.83 and mean GOHAI was 5.79. Both measures disclosed the impacts of oral problems in the assessed population, but the GOHAI exposed an issue that was not revealed by the OHIP-14. Interestingly, the OHIP-14 was not associated with medical data like diabetes mellitus, myocardial infraction, and hypertension. Moreover, OR analysis did not confirm the association between diabetes and hypertension with either scale describing OHRQoL. According to Gerritsen et al., the shortened version of OHIP-14 can lead to under-reporting impacts, a further reason for performing two measures instead of one [18]. Some of the issues reported in OHRQoL studies may vary due to the evaluated variables, for example, participants’ age, the affluence of the society, or the educational level.

Tooth loss is associated with unfavorable oral health–related quality of life (OHRQoL), which increases severely when the number of preserved teeth decreases below 20 [18]. Most respondents in our study (almost 66%) had more than 20 teeth; 20 teeth has been considered the threshold for proper mastication [30]. Both measures showed significant association with number of teeth. Missing teeth (MT) were related with all the variables: gender, age, dry mouth, education, professional status, BMI, diabetes, myocardial infraction, hypertension, number of teeth, and total dentures in lower and/or upper jaws. This proves again that missing teeth is a significant risk factor, not only in regard to demographic data but also medical data like obesity, diabetes mellitus, myocardial infraction, and hypertension. Interestingly, in our study edentulous individuals had a two times higher risk of having only a OHIP-14 score above the median. Moreover, there is a visible association with OHRQoL, but individual measures do not necessary report unsatisfactory results. Therefore, it is not enough to perform only the OHIP-14 or GOHAI. They have to be compared with objective socio-economic factors, medical data, and dental examination. 

In our study “pain and discomfort” was the most often reported measure in both the GOHAI and OHIP-14 measure. In the GOHAI measure, almost half of the participants (48.3%) experienced sensitive teeth and gums to hot, cold, or sweetness, and in OHIP-14 one quarter (25.3%) found it uncomfortable to eat any food. A plausible explanation may be the presence of dental hypersensitivity. Generally, its prevalence rate ranges between 3% to even 98% in different populations, with age peak around 30–40 years [31]. According to Bekes et al., hypersensitive teeth are significantly associated with impaired OHRQoL [32]. It may be that the pain and discomfort of sensitive teeth were reflected in other responses in the psychological impact domain (concerned or worried and tense or nervous) as well as in functional limitations (trouble biting/chewing food). In this case as well, it was the GOHAI that disclosed issues that were less reflected by the other scale [21]. GOHAI is aimed to perform well in older populations, but our study showed that it is a reliable tool in the adult population as well.

OHRQoL, among our patients, was satisfying but we do not know how patients prioritize their oral health. Dental caries and periodontal disease, the main causes of tooth loss in adults, mostly develop as chronic diseases. Over time, people can accept their deteriorating oral health condition through, for example, slowly changing their food choices or social activities [33]. 

We believe that our study will help clinicians and policymakers better understand the relationship between clinical data regarding oral health (dental caries, tooth loss) and general health and the subjective evaluation of oral health–related quality of life measures. We suggest using both scales simultaneously in a population with a wide age range. This makes it possible to assess the entire spectrum of oral health problems in relation to quality of life.

## 5. Conclusions

In our study, even though the respondents were not a senior group, we detected a significant relationship between oral health–related quality of life and factors influencing the presence or absence of dentition. Missing teeth were statistically associated with GOHAI, OHIP-14, advanced age, self-reported dry mouth, lower education, higher BMI, lower professional status, diabetes, myocardial infraction, and total dentures in upper or/and lower jaws. The subjective OHRQoL measures should be compared with objective socio-economic factors, medical data, and dental examination to present the whole spectrum of problems and the interaction between targeted conditions and psychosocial approaches of the participant.

## Figures and Tables

**Table 1 ijerph-19-01626-t001:** Response percentages for very often, fairly often, and occasionally in GOHAI and OHIP-14 questionnaire.

GOHAI %	OHIP-14 %
Functional limitation	
Trouble biting/chewing food 21.9	Trouble pronouncing words 6.8
Uncomfortable to swallow 3.3	Sense of taste worse 7.2
Prevented from speaking 9.0	
Pain and discomfort	
Discomfort when eating 5.6	Painful aching in mouth 12.8
Use medication to relieve pain 12.0	Uncomfortable to eat foods 25.1
Teeth, gums sensitive to hot/cold 48.0	
Psychological impacts	
Unhappy with appearance 12.9	Been self-conscious 16.2
Worried or concerned 42.0	Felt tense 18.8
Nervous or self-conscious 24.7	Difficult to relax 12.9
Uncomfortable eating in front of people 13.3	Been embarrassed 9.1
	Felt life is less satisfying 9.4
Behavioral impacts	
Limit kinds or amounts of food 13.8	Diet has been unsatisfactory 16.1
Limit contact with others 4.3	Had to interrupt meals 9.7
	Been irritable with others 4.0
	Difficulty doing usual jobs 14.8
	Totally unable to function 2.4

GOHAI—Geriatric/General Oral Health Assessment Index; OHIP-14—Oral Health Impact Profile.

**Table 2 ijerph-19-01626-t002:** Mean values of OHIP-14 and GOHAI and DMFT scores of selected variables.

	GOHAI n/Mean	p	OHIP-14 n/Mean	p	DT	p	MT	p	FT	p	DMFT	p
Total	1112(5.8)	0.000	1112(2.8)	0.005	1112(0.6)	0.000	1112(6.1)	0.000	1112(8.8)	0.000	1112(15.5)	0.000
Gender M	502(4.9)		502(2.3)		502(0.9)		502(5.4)		502(8.2)		502(14.5)	
W	610(6.5)		610(3.3)		610(0.4)		610(6.6)		610(9.4)		610(16.4)	
Age ˂65	891(5.4)	0.000	891(2.6)	0.000	891(0.7)	0.0049	891(3.9)	0.000	891(9.9)	0.000	891(14.6)	0.000
65+	221(7.3)		221(3.8)		221(0.4)		221(14.5)		221(4.5)		221(19.4)	
Dry mouth no	866(5.1)	0.000	866(2.4)	0.000	866(0.6)	0.908	866(5.3)	0.000	866(9.2)	0.001	866(15.1)	0.000
yes	246(8.1)		246(4.3)		246(0.6)		246(8.9)		246(7.7)		246(17.2)	
Education elementary	36(9.5)	0.000	36(5.6)	0.000	36(1.2)	0.001	36(12.5)	0.000	36(4.8)	0.000	36(18.5)	0.000
secondary	511(6.3)		511(3.4)		511(0.8)		511(8.5)		511(7.4)		511(16.7)	
higher	565(5.0)		565(2.1)		565(0.4)		565(3.5)		565(10.4)		565(14.3)	
BMI ≤25	430(5.9)	0.004	430(2.6)	0.096	430(0.6)	0.259	430(4.0)	0.000	430(9.9)	0.000	430(14.6)	0.000
25–30	402(5.1)		402(2.6)		402(0.5)		402(6.1)		402(8.9)		402(15.6)	
>30	280(6.6)		280(3.4)		280(0.7)		280(9.2)		280(7.0)		280(16.9)	
Professinal status working	725(5.0)	0.000	725(2.3)	0.000	725(0.6)	0.027	725(3.5)	0.000	725(10.3)	0.000	725(14.4)	0.000
annuity	18(10.7)		18(8.6)		18(1.2)		18(8.6)		18(8.2)		18(18.1)	
unemployed	99(7.2)		99(3.8)		99(1.1)		99(3.6)		99(8.8)		99(13.5)	
pension	270(6.9)		270(3.4)		270(0.4)		270(13.8)		270(4.9)		270(19.2)	
Diabetes mellitus no	1035(5.6)	0.009	1035(2.7)	0.135	1035(0.6)	0.616	1035(5.5)	0.000	1035(9.1)	0.000	1035(15.2)	0.000
yes	77(8.2)		77(4.5)		77(0.6)		77(13.5)		77(5.4)		77(19.5)	
Myocardial infraction no	1090(5.7)	0.088	1090(2.7)	0.018	1090(0.6)	0.379	1090(5.9)	0.000	1090(8.9)	0.000	1090(15.4)	0.001
yes	22(9.5)		22(6.2)		22(0.5)		22(15.9)		22(4.0)		22(20.4)	
Hypertension no	793(5.4)	0.004	793(2.5)	0.054	793(0.6)	0.554	793(4.3)	0.000	793(9.6)	0.000	793(14.6)	0.000
yes	319(6.6)		319(3.6)		319(0.5)		319(10.5)		319(6.9)		319(17.9)	
Number of teeth ˂20	379(7.8)	0.000	379(4.6)	0.000	379(0.5)	0.240	379(14.3)	0.000	379(4.4)	0.000	379(19.2)	0.000
20+	733(4.7)		733(1.9)		733(0.6)		733(1.8)		755(11.1)		733(13.6)	
Total dentures upper jaw no	1038(5.5)	0.000	1038(2.6)	0.000	1038(0.6)	0.015	1038(4.7)	0.000	1038(9.4)	0.000	1038(14.8)	0.000
yes	74(9.2)		74(5.6)		74(0.2)		74(24.6)		74(0.5)		74(25.3)	
Total dentures lower jaw no	1060(5.6)	0.001	1060(2.7)	0.000	1060(0.6)	0.000	1060(5.1)	0.000	1060(9.3)	0.000	1060(15.0)	0.000
yes	52(8.4)		52(5.7)		52(0.0)		52(25.3)		52(0.4)		52(25.8)	

GOHAI—Geriatric/General Oral Health Assessment Index; OHIP-14—Oral Health Impact Profile; DT—decay teeth; MT—missing teeth; FT—filled teeth; DMFT—decay, missing teeth, filled teeth; BMI—Body Mass Index.

**Table 3 ijerph-19-01626-t003:** Spearman’s correlation coefficient between GOHAI and OHIP-14 scores (r-Spearman’s correlation coefficient).

	GOHAI		OHIP-14	
	r	p	r	p
Age	0.19	0.000	0.15	0.000
DT	0.12	0.000	0.10	0.001
MT	0.26	0.000	0.23	0.000
FT	−0.13	0.000	−0.14	0.000
DMFT	0.25	0.000	0.21	0.000

GOHAI—Geriatric/General Oral Health Assessment Index; OHIP-14—Oral Health Impact Profile; DT—decay teeth; MT—missing teeth; FT—filled teeth; DMFT—decay, missing teeth, filled teeth.

**Table 4 ijerph-19-01626-t004:** Univariable logistic regression of oral health status and selected diseases in relation to OHIP-14 and GOHAI scores above the median value.

Independent Variables	GOHAI > 4			OHIP-14 > 0		
	p	OR	CI 95%	p	OR	CI 95%
* Number of teeth	0.000	0.958	0.944–0.972	0.000	0.960	0.947–0.974
Edentulousness	0.081	1.758	0.932–3.314	0.018	2.171	1.142–4.128
Total denture upper jaw	0.000	2.489	1.499–4.134	0.000	2.434	1.481–4.001
Total denture lower jaw	0.015	2.068	1.153–3.708	0.002	2.516	1.392–4.548
Dry mouth	0.000	2.547	1.894–3.426	0.004	1.512	1.138–2.010
Diabetes mellitus	0.120	1.449	0.908–2.313	0.401	1.219	0.767–1.938
Myocardial infraction	0.571	1.277	0.547–2.981	0.042	2.557	1.034–6.320
Hypertension	0.083	1.259	0.971–1.634	0.280	1.154	0.890–1.498

* Independent variable treated as continuous in logistic regression model. GOHAI—Geriatric/General Oral Health Assessment Index; OHIP-14—Oral Health Impact Profile.

## Data Availability

Data are available on request from the authors.

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
