# Peer review of "Oral Health–Related Quality of Life and Missing Teeth in an Adult Population: A Cross-Sectional Study from Poland"

_ijerph, 2022, doi:10.3390/ijerph19031626_

Round 1

Reviewer 1 Report

This is an interesting paper aimed to determine the oral health related quality of life (OHRQoL)

in relation to missing teeth in Polish adult population aged 20–79 years. Authors have used two patient reported outcome measures GOHAI and OHIP-14 for evaluating the OHRQoL in this study.  

In the abstract section, authors have mentioned using the random sampling while it was not presented in the method section. In addition, the study hypothesis as well as rationale for conducting the current study is missing in the introduction section.

How many clinical examiners registered the oral findings? Were they calibrated? The duration of study is relatively long, were they recalibrated? How were information on study covariates collected? How was odds ratio computed?

Why do authors have two decimals in mean values and one in the proportion? However, in the text it is different. I suggest revising these values into 1 decimal place. Furthermore, authors have compared the Mean/proportion between groups and reported the result as “significantly associated” based on the p-values generated from tests. The same with the “Odds Ratio”. I suggest removing these terms throughout the text in the result section. Furthermore, I recommend reading the following paper: Rafi & Greenland (2020): Semantic and cognitive tools to aid statistical science: replace confidence and significance by compatibility and surprise. https://doi.org/10.1186/s12874-020-01105-9. Please check the caption of Table 4. In addition, ORs are just >1 for all covariates, how would authors explain these findings?  

The discussion section should be revised and restructured. I suggest reading Docherty M, Smith R. The case for structuring the discussion of scientific papers. BMJ. 1999;318(7193):1224-1225. This paper lacks clinical implications, and generalisability.

I recommend language proofreading. 

Author Response

This is an interesting paper aimed to determine the oral health related quality of life (OHRQoL)

in relation to missing teeth in Polish adult population aged 20–79 years. Authors have used two patient reported outcome measures GOHAI and OHIP-14 for evaluating the OHRQoL in this study.

In the abstract section, authors have mentioned using the random sampling while it was not presented in the method section. – we have added such information in the method section.

In addition, the study hypothesis as well as rationale for conducting the current study is missing in the introduction section. – we have added the hypothesis and the rationale of the study in the introduction section.

How many clinical examiners registered the oral findings? - 4, such information has been added in the text.

 Were they calibrated? Yes, such information has been added in the text.

 The duration of study is relatively long, were they recalibrated? No, there was no recalibration. But this is a very valuable remark, and we will provide recalibration. Thank you for this valuable comment.

How were information on study covariates collected? The information was collected using a questionnaire.

How was odds ratio computed? - The odds ratio was computed to present the results of logistic regression model and the relevant information has been added in statistical analysis

Why do authors have two decimals in mean values and one in the proportion? – we have corrected it to one decimal as suggested

However, in the text it is different. I suggest revising these values into 1 decimal place. – We have corrected it to one decimal.

 Furthermore, authors have compared the Mean/proportion between groups and reported the result as “significantly associated” based on the p-values generated from tests. -  Thank you very much for this article. It was very interesting; however, we have decided to choose well known terminology that have been used and understood for many years. We do not want to confuse the terminology and the readers, especially as the articles we cited used the same terminology as we did.

 The same with the “Odds Ratio”. I suggest removing these terms throughout the text in the result section. – We have done as suggested.

 Furthermore, I recommend reading the following paper: Rafi & Greenland (2020): Semantic and cognitive tools to aid statistical science: replace confidence and significance by compatibility and surprise. https://doi.org/10.1186/s12874-020-01105-9. Thank you for this recommendation. We have read this article.

Please check the caption of Table 4. In addition, ORs are just >1 for all covariates, how would authors explain these findings?  - The table 4 was checked once again and findings were described under the table

The discussion section should be revised and restructured.- We have read the suggested article, revised and restructured the discussion section.

 I suggest reading Docherty M, Smith R. The case for structuring the discussion of scientific papers. BMJ. 1999;318(7193):1224-1225. We have read this article, thank you very much

 This paper lacks clinical implications, and generalisability. – We have added, in the discussion section, information about clinical implications and generalisability

I recommend language proofreading. This paper has been proofread again by a professional English translator.

Reviewer 2 Report

I would like to congratulate the authors for their wonderful work.

However I find the results not very explicit. Tables 1 and 2 are difficult to understand and follow and the explanations are very brief.

Also, I would consider mor specific conclusions to be better suited.

Author Response

I would like to congratulate the authors for their wonderful work.

However I find the results not very explicit.

 Tables 1 and 2 are difficult to understand and follow and the explanations are very brief.- We have explained table 1 and corrected table 2 to be more clear.

Also, I would consider mor specific conclusions to be better suited. We have written more specific conclusions.

Thank you for the very in-depth review

Reviewer 3 Report

Dear authors,

Thank you very much for your paper. In this paper, the authors presented a study entitled “Oral Health Related Quality of Life and Missing Teeth in Adult Populational from Poland”

aiming to s to determine oral health related quality of life (OHRQoL) using both measures GOHAI and OHIP-14 in relation to missing teeth in Polish population age 20-79.

In general, the manuscript is very interesting and well-written. However, minor  corrections are required to improve the overall quality. An English-language review is required.

My recommendations are the following:

Please insert on the title and abstract the type of the study in order to be immediately understandable for the reader. 

The abstract section is well-performed . The introduction section should be improved. Please describe the situation in other countries.

Discussion

  1. Please consider to begin this section accepting or rejecting the null hypothesis.
  2. Add limitations of the study in a separate paragraph

Author Response

Thank you very much for your paper. In this paper, the authors presented a study entitled “Oral Health Related Quality of Life and Missing Teeth in Adult Populational from Poland”

aiming to s to determine oral health related quality of life (OHRQoL) using both measures GOHAI and OHIP-14 in relation to missing teeth in Polish population age 20-79.

In general, the manuscript is very interesting and well-written. However, minor corrections are required to improve the overall quality. An English-language review is required.- We asked a professional English translator to proofread the manuscript.

My recommendations are the following:

Please insert on the title and abstract the type of the study in order to be immediately understandable for the reader.  – It has been inserted.

Thank you for the very in-depth review

The abstract section is well-performed . The introduction section should be improved. Please describe the situation in other countries. – We have added information about the situation in other countries.

Discussion

  1. Please consider to begin this section accepting or rejecting the null hypothesis. – We restructured discussion section accordingly

2 Add limitations of the study in a separate paragraph- Limitations of the study have been presented in a new paragraph.

Round 2

Reviewer 1 Report

How was randomization carried out using the Bialystok Mayor’s Office database?Authors have mentioned that 3246 residents were invited for the study and only 1196 participated.

Authors have mentioned about the use of questionnaire for investigating the covariates. These questions must be opened-up. The present questions on BMI, DM, MI, Hypertension without clinical measurement seems to be interesting for readers.

Authors have mentioned that logistic regression models were used to compute OR in the present study. Did authors dichotomize the outcome variable? If yes, how did authors dichotomize? Did authors present the crude odds?

Author Response

Thank you very much for your insightful analysis of our work and your valuable comments, which allowed us to correct the results included in the paper and improve the manuscript significantly.

How was randomization carried out using the Bialystok Mayor’s Office database? Authors have mentioned that 3246 residents were invited for the study and only 1196 participated. - Bialystok Mayor Office provided an anonymous list of residents of Bialystok city. Random sample of residents were chosen using random number generator (based on Mersenne twister algorithm). Until the analysis was carried out 3246 residents were invited. But only 1196 decided to participate.

Authors have mentioned about the use of questionnaire for investigating the covariates. These questions must be opened-up. The present questions on BMI, DM, MI, Hypertension without clinical measurement seems to be interesting for readers. -Information on diseases were based on interview questionnaire. BMI was assessed based on participants body weight and height measurements according to the formula BMI=body weight in kg/height in m2

Authors have mentioned that logistic regression models were used to compute OR in the present study.

Did authors dichotomize the outcome variable? If yes, how did authors dichotomize? Values of OHIP-14 and GOHAI scales for the purpose of using logistic regression were dichotomized based on median value.

Did authors present the crude odds? Odd ratio presented in table 4 come from univariable logistic regression model

Reviewer 2 Report

I consider the paper to be improved significantly and I recommand publication.

Author Response

Thank you very much for your valuable comments and time you have spent reading our paper.